# Interplay between the Glymphatic System and the Endocannabinoid System: Implications for Brain Health and Disease

**DOI:** 10.3390/ijms242417458

**Published:** 2023-12-14

**Authors:** Juan F. Osuna-Ramos, Josué Camberos-Barraza, Laura E. Torres-Mondragón, Ángel R. Rábago-Monzón, Alejandro Camacho-Zamora, Marco A. Valdez-Flores, Carla E. Angulo-Rojo, Alma M. Guadrón-Llanos, Verónica J. Picos-Cárdenas, Loranda Calderón-Zamora, Javier A. Magaña-Gómez, Claudia D. Norzagaray-Valenzuela, Feliznando I. Cárdenas-Torres, Alberto K. De la Herrán-Arita

**Affiliations:** 1Faculty of Medicine, Autonomous University of Sinaloa, Culiacán 80019, Mexico; 2Doctorado en Biomedicina Molecular, Autonomous University of Sinaloa, Culiacán 80019, Mexico; 3Maestría en Biomedicina Molecular, Autonomous University of Sinaloa, Culiacán 80019, Mexico; 4Faculty of Biology, Autonomous University of Sinaloa, Culiacán 80019, Mexico; 5Faculty of Nutrition Sciences and Gastronomy, Autonomous University of Sinaloa, Culiacán 80019, Mexico

**Keywords:** GS, endocannabinoid system, waste clearance, neurodegenerative disorders, brain health

## Abstract

The intricate mechanisms governing brain health and function have long been subjects of extensive investigation. Recent research has shed light on two pivotal systems, the glymphatic system and the endocannabinoid system, and their profound role within the central nervous system. The glymphatic system is a recently discovered waste clearance system within the brain that facilitates the efficient removal of toxic waste products and metabolites from the central nervous system. It relies on the unique properties of the brain’s extracellular space and is primarily driven by cerebrospinal fluid and glial cells. Conversely, the endocannabinoid system, a multifaceted signaling network, is intricately involved in diverse physiological processes and has been associated with modulating synaptic plasticity, nociception, affective states, appetite regulation, and immune responses. This scientific review delves into the intricate interconnections between these two systems, exploring their combined influence on brain health and disease. By elucidating the synergistic effects of glymphatic function and endocannabinoid signaling, this review aims to deepen our understanding of their implications for neurological disorders, immune responses, and cognitive well-being.

## 1. Introduction

The intricate and dynamic nature of the central nervous system (CNS) involves a multitude of regulatory systems that contribute to its structural integrity, functional balance, and resilience against pathological insults. Among these, the Glymphatic System (GS) and the Endocannabinoid System (ECS) have emerged as crucial players, each exerting profound influences on neurological processes. The GS, a relatively recent addition to our understanding of CNS physiology, functions as a waste clearance mechanism, facilitating the removal of metabolites and potentially neurotoxic substances. In parallel, the ECS, a complex signaling network, modulates neurotransmission, immune responses, and inflammatory processes throughout the brain.

As research progresses, an intriguing intersection between the GS and the ECS has become increasingly apparent, suggesting a sophisticated interplay that extends beyond their functions. This scientific review seeks to comprehensively explore the intricate relationship between the GS and the ECS, investigating how their coordination may impact the broader landscape of brain health and disease. By unraveling the molecular and cellular dialogues between these systems, we aim to elucidate their collective influence on key aspects such as neuroinflammation, blood-brain barrier regulation, and overall CNS homeostasis.

Understanding the nuanced interactions between the GS and the ECS holds substantial promise for uncovering novel therapeutic avenues and refining our approach to managing neurological disorders. This review provides a comprehensive overview of the current state of knowledge regarding the interplay between the GS and the ECS, offering insights into potential mechanisms, implications for neurological diseases, and avenues for future research.

## 2. The Glymphatic System

The brain is a highly metabolically active organ that generates substantial waste products that require efficient clearance mechanisms. The GS, initially described by Iliff et al. in 2012, represents a novel waste clearance pathway within the CNS [1]. It involves the exchange of cerebrospinal fluid (CSF) and interstitial fluid (ISF) facilitated by a complex network of interconnected structures within the brain (Figure 1). Key anatomical components include the perivascular spaces surrounding arteries and veins, aquaporin-4 (AQP4) channels on astrocytic endfeet, and the meningeal lymphatic vessels. The perivascular spaces serve as conduits for the convective flow of CSF, while astrocytic AQP4 channels regulate the influx and efflux of fluids [2,3] (Figure 1, inset A). In addition, the recently discovered meningeal lymphatic vessels provide a route for waste clearance from the CNS to the peripheral lymphatic system [4,5].

The GS operates through a combination of convective bulk flow and diffusion along concentration gradients. CSF enters the brain parenchyma through the perivascular spaces, exchanging waste products with the ISF. This dynamic process facilitates the efficient clearance of metabolic waste, including β-amyloid (Aβ), tau proteins, neurotransmitters, and other solutes from the brain. The GS also participates in the regulation of ion and neurotransmitter balance, nutrient delivery, and the modulation of neural activity [2,3,4,5].

The GS is subject to dynamic regulation by various factors. Sleep plays a crucial role in modulating glymphatic function, with increased glymphatic activity occurring during non-rapid eye movement stage 3 (NREM 3). Arousal level, blood-brain barrier (BBB) permeability, and neurotransmitters such as norepinephrine, adenosine, and gamma-aminobutyric acid (GABA) influence glymphatic clearance efficiency. The GS is also influenced by vascular pulsatility, glial cell activity, and the glymphatic-lymphatic interaction [6,7,8].

Impairments in glymphatic function have been implicated in several neurological disorders. Accumulation of Aβ and tau proteins, as well as other metabolic waste, due to compromised glymphatic clearance mechanisms, is thought to contribute to the pathogenesis and progression of Alzheimer’s disease (AD), amyotrophic lateral sclerosis (ALS), Parkinson’s disease (PD), multiple sclerosis (MS), among others [9].

Nowadays, it is feasible to evaluate glymphatic function using Diffusion Tensor Imaging (DTI). This is a magnetic resonance imaging (MRI) technique that measures the diffusion of water molecules in tissues. It is commonly used to investigate the microstructure of white matter tracts in the brain by assessing the directionality of water diffusion. By using specialized imaging protocols to study water diffusion and structural characteristics around blood vessels, it is possible to independently measure diffusion within the perivascular spaces of the medullary arteries and veins apart from diffusion along the larger white matter fibers. An assessment methodology known as Diffusion Tensor Imaging along the Perivascular Space (DTI-ALPS) is used to assess GS activity.

With the DTI-ALPS technique, the diffusion capacity within the perivascular space in the white matter on the outer aspect of the lateral ventricle’s body is assessed as a ratio concerning the diffusion capacity within the perivascular space and that in a direction perpendicular to the principal flow direction of the white matter fibers (referred to as the ALPS index) [5,10,11,12,13,14,15,16,17,18,19,20].

Accurately modeling the GS has the potential to yield valuable novel insights into GS dynamics and pathways, ultimately enabling the creation of quantitative maps that can be used for disease diagnosis, monitoring, and prognosis [18,20,21,22,23,24].

## 3. The Blood-Brain Barrier

The BBB is a specialized barrier formed by endothelial cells in brain capillaries. It serves as a physical and functional barrier, tightly regulating the exchange of substances between the blood and the brain.

The unique characteristics of the brain endothelium enable a critical limitation through the absence of fenestrations and the presence of tight junctions between cells. The architectural composition of the BBB consists of endothelial cells that line the microvessels of the brain, capillary basement membranes, and specialized structures known as end-feet, which extend from astrocytes to the basement membrane [17,25].

The selective permeability of the BBB is ensured by intercellular tight junctions that connect brain microvascular endothelial cells. The blood vessels of the BBB are lined with endothelial cells, which are tightly interconnected through tight junctions. Surrounding the basement membrane, astrocyte endfeet and pericytes contribute to the structural support of the BBB (Figure 1, inset A). The main function of tight junctions is to restrict the paracellular pathway for the diffusion of hydrophilic solutes, thereby allowing for the control of chemical substances present in the circulatory system that can access the brain [26,27].

These features of brain vessels regulate the exchange of molecules and cells between the circulation and the CNS. Additionally, they play crucial roles in preventing the loss of essential substances by controlling transcellular movements of water, ions, oxygen, carbon dioxide, and glucose, all of which are necessary for cerebral cellular metabolism. Consequently, disorders affecting the BBB can disrupt ion regulation and disturb homeostasis, leading to impaired brain function [27,28].

Given that glymphatic dysfunction can lead to the accumulation of waste products in the brain and contribute to the genesis of neurodegenerative diseases [29], the integrity and functionality of the BBB are crucial for the clearance of these waste products from the brain parenchyma. Moreover, the BBB also controls the entry of nutrients and oxygen into the brain, providing essential energy substrates for neuronal function and maintenance. On the other hand, the GS, by facilitating the exchange of CSF and ISF, ensures the efficient distribution of these nutrients and oxygen to different regions of the brain. Disruptions in either the GS or the BBB can lead to inadequate nutrient supply and subsequent neuronal dysfunction [24,25,28].

In addition, both the GS and the BBB contribute to the regulation of the brain’s immune response. Dysfunction in either system can lead to increased inflammation and impaired clearance of inflammatory molecules, which can negatively impact brain health. Research suggests that an interplay between the GS and the BBB is crucial for the efficient clearance of immune cells and inflammatory mediators from the brain [30,31].

The relationship between the GS and the BBB is vital for maintaining brain homeostasis. Their coordination ensures the efficient removal of waste products, proper nutrient supply, and regulation of the immune response [24,32].

## 4. The Endocannabinoid System

The ECS is a complex signaling network present throughout the body. The ECS was first discovered through research on the pharmacological effects of cannabis-derived compounds. It is now recognized as a crucial regulatory system involved in maintaining homeostasis throughout the body. The ECS comprises endogenous cannabinoids (endocannabinoids), cannabinoid receptors (CB1R and CB2R), and enzymes responsible for endocannabinoid synthesis and degradation [33,34].

The two primary endocannabinoids are anandamide (AEA) and 2-arachidonoylglycerol (2-AG), whereas enzymes involved in endocannabinoid metabolism include fatty acid amide hydrolase (FAAH) and monoacylglycerol lipase (MAGL), which are responsible for the degradation of AEA and 2-AG, respectively [35,36].

Unlike conventional neurotransmitters, endocannabinoids are not stored in vesicles but are synthesized by neurons on demand, utilizing lipid components of the cell membrane. These endocannabinoids function as retrograde messengers, transmitting intercellular signals from postsynaptic neurons back to presynaptic terminals, where they inhibit the release of neurotransmitters (Figure 1, inset A) [35,36,37].

Initially, their pharmacological properties were thought to be similar, with the assumption that these molecules could be interchangeably and indistinguishably involved in regulating synaptic functions, synaptic plasticity, and behavioral aspects such as learning, memory, reward, addiction, antinociception, and anxiety. However, emerging evidence suggests that AEA and 2-AG possess distinct pharmacological properties, contribute to different forms of synaptic plasticity, and participate in diverse behavioral functions [33,37,38,39].

Apart from endocannabinoids, phytocannabinoids constitute a diverse array of compounds with the capacity to activate cannabinoid receptors. Cannabidiol (CBD) and tetrahydrocannabinol (THC) stand as the most extensively studied phytocannabinoids originating from the cannabis plant. They display an extensive spectrum of therapeutic potential, attracting substantial attention due to their interactions with the body’s endocannabinoid receptors. In addition to CBD and THC, alternative phytocannabinoids, like cannabigerol (CBG) and cannabinol (CBN), also engage cannabinoid receptors. These interactions incite a myriad of physiological responses, influencing processes such as nociception, immune modulation, and inflammatory regulation, among others [33,34,35,36,37].

These endogenous and exogenous cannabinoids bind to and activate cannabinoid receptors, predominantly CB1R and CB2R, which are widely distributed throughout the body. The human body possesses specific binding sites for cannabinoids that are distributed across the surfaces of various cells. These receptors are part of the extensive family of G protein-coupled receptors (GPCRs), which encompasses the majority of commonly found receptors. GPCRs are membrane receptors composed of seven transmembrane domains (7TM), featuring an extracellular amino-terminal and an intracellular carbonyl terminal [36,37].

Upon ligand binding, the activated cannabinoid receptors interact with and activate specific G proteins located inside the cell. CB1R predominantly activates Gαi/o proteins, while CB2R can couple with Gαi/o, as well as Gαs proteins.

The activated G proteins dissociate into their subunits, leading to the modulation of intracellular signaling pathways. In the case of CB1R, Gαi/o inhibits the production of cyclic adenosine monophosphate (cAMP) and reduces calcium ion influx into neurons, thereby modulating neurotransmitter release. CB2R activation can also inhibit cAMP production but stimulates other signaling pathways involved in immune responses and inflammation, like the MAP kinase phosphatase 3 (MKP3) pathway. The primary function of cannabinoid receptors is to regulate the release of other chemical messengers. CB1R modulates the release of specific neurotransmitters, thereby protecting the CNS from excessive stimulation or inhibition caused by other neurotransmitters. On the other hand, CB2R primarily plays a peripheral role via immunomodulatory activities, primarily modulating the release of cytokines, protein molecules responsible for regulating immune function, and inflammatory responses [36] (Figure 1, inset B). Consequently, cannabinoids may impact neurodegenerative diseases through their neuro- and immunomodulatory effects [40].

Cannabinoid receptors exhibit distinct tissue distributions and signaling mechanisms. Among them, CB1Rs are highly abundant and widely distributed within the brain. They are primarily located on neurons in the CNS. In the brain, CB1Rs are particularly prominent in regions associated with motor coordination and movement (e.g., cerebellum, basal ganglia, striatum, substantia nigra), attention, and complex cognitive functions like judgment (e.g., cerebral cortex), learning, memory, and emotions (e.g., amygdala, hippocampus). CB1Rs are also present, albeit to a lesser extent, in select organs and peripheral tissues, including endocrine glands, salivary glands, leukocytes, spleen, heart, and parts of the reproductive, urinary, and gastrointestinal systems. In addition, the activation of CB1R in astrocytes, whether prompted by endogenous or exogenous cannabinoids, initiates intracellular signaling resulting in an elevation of cytosolic calcium. This, in consequence, prompts the release of glutamate from astrocytes (Figure 1, inset B). The released glutamate then activates extra-synaptic NMDA receptors in neurons located at a considerable distance, giving rise to depolarizing currents termed slow inward currents (SICs). Various investigations have demonstrated the involvement of SICs in neuronal synchronization [37].

The distribution of CB1R suggests that endocannabinoids play a physiological role in the regulation of movement and sensory perception, as well as in processes related to learning, memory, and emotional states such as pleasure and aggression. In contrast to CB1Rs, CB2Rs are primarily expressed in immune cells and tissues, such as leukocytes, spleen, tonsils, bone marrow, and to a lesser extent, in the pancreas. Recent findings have also identified CB2Rs in the CNS, specifically on glial and microglial cells, where they induce a reduction of pro-inflammatory factors and microglial migration via MKP-3 (Figure 1, inset B) [37,41].

The expression of CB1R and CB2R in microglia is contingent upon their activation state and phenotype. Activated microglia in brain tissue express CB2R; however, the specific neuropathology influencing their activation results in varying phenotypes and levels of CB2R expression. Activation of microglial CB2R through cannabinoids plays a regulatory role in their immune-related functions. For instance, CB2R activation enhances microglial cell proliferation and migration, concurrently decreasing the release of harmful factors like TNFα and free radicals. Given the myriad of tissues and cells that express cannabinoid receptors, the ECS plays a vital role in various physiological processes, including pain modulation, immune function, inflammation, appetite regulation, metabolism, neuronal plasticity, and stress responses. It also acts as a homeostatic regulator, maintaining balance within the body [36,37,38,39,40,41].

Dysregulation of the ECS has been implicated in numerous health conditions. Alterations in endocannabinoid signaling have been associated with chronic pain, inflammatory disorders, neurodegenerative diseases, metabolic disorders, mood disorders, and addiction [40,41].

## 5. The Endocannabinoid System and the Blood-Brain Barrier

Inflammation within the CNS is one of the main mechanisms involved in the development of neurological conditions. This process is initiated as a complex cascade triggered by inflammatory signals arising from infection, injury, or neurodegenerative processes, and it involves the orchestrated migration of leukocytes, such as neutrophils and monocytes, from the bloodstream through BBB to the site of inflammation. Upon reaching the BBB, leukocytes engage in a series of steps, including initial rolling and subsequent firm adhesion to endothelial cells, followed by transmigration through the BBB and into the CNS tissue. Within the CNS tissue, these migrating leukocytes interact with local immune cells, particularly microglia, resulting in the activation of an immune response. This immune response entails the release of pro-inflammatory cytokines, chemokines, and other mediators, aiming to eliminate the source of inflammation. The interplay between circulating leukocytes and resident immune cells can amplify the immune response, leading to an intensified inflammatory process [30,31]. A balanced resolution of inflammation is crucial to prevent tissue damage and neurodegenerative disorders [42]. Inflammatory conditions and neurodegenerative diseases disrupt the BBB, leading to increased permeability and compromised brain function.

The ECS may exert regulatory effects on the BBB, potentially contributing to the maintenance of barrier integrity under pathological conditions. Consequently, there has been a growing interest in cannabinoids for their anti-inflammatory and immunomodulatory properties, as well as their capacity to modulate endothelial and epithelial barriers, making them promising candidates for improving cognitive deficits by protecting the BBB.

The ECS components have been identified in the cells of the BBB, including endothelial cells and astrocytes. CB1Rs are primarily located on the luminal side of the BBB endothelium. Expression of CB1R has been observed in astrocytes, microglial cells, and pericytes. CB2Rs, on the other hand, are located on the abluminal side of the BBB (Figure 1, inset A). Activation of CB1R in astrocytes regulates their metabolic activities. For instance, when CB1R on astrocytes is activated, it enhances the rates of both glucose oxidation and ketogenesis, two processes crucial for supplying energy to the brain. Taking into consideration the capacity of astrocytic CB1R to oversee energy metabolism and facilitate neuron-glia interactions, one could conjecture their potential pathological significance. Considering the pivotal role of perivascular astrocytes in delivering energy from the blood to neurons in an activity-dependent fashion, it is plausible that astrocytic CB1R plays a role in regulating the energy supply to neurons [43,44].

Building on the influence of cannabinoid receptor activation on endothelial cell responses, further insights emerge from investigations into cannabinoid modulation. Through the use of an experimental model for multiple sclerosis known as Theiler’s murine encephalomyelitis virus-induced demyelinating disease, researchers have demonstrated that CBD can modify the detrimental effects of this condition by reducing the infiltration of leukocytes from the systemic circulation. This effect is achieved through down-regulating the expression of chemokines such as C-C motif chemokine ligand 2 (CCL2) and C-C motif chemokine ligand 5 (CCL5), interleukin-1 β, and VCAM-1 [45]. This suggests that the ECS mitigates the observed BBB alterations in these conditions by preventing inflammation in endothelial cells. The authors also found that CBD can prevent cellular margination and reduce the expression of adhesion molecules and chemotaxis in laboratory models of inflammation.

In a similar study, CBD administration enhanced the integrity and permeability of the BBB and diminished the protein levels of proinflammatory cytokines (TNFα and IL-1β). Additionally, it substantially elevated the expression of tight junction proteins (claudin-5 and occludin) [46].

Related research has yielded further insights into the potential of cannabinoids in managing neuroinflammatory processes. CB2R activation has been shown to decrease the induction of intercellular adhesion molecule-1 (ICAM-1) and VCAM-1 surface expression and increase the amount of tight junction proteins in human brain microvascular endothelial cells exposed to various proinflammatory mediators [47]. In addition, CB2R activation has also been shown to attenuate BBB damage and neurodegeneration in a murine model of traumatic brain injury through the modulation of macrophage/microglia cell response [48].

The ECS demonstrates immunomodulatory properties that could potentially impact immune responses within the brain, particularly pertinent in the context of neuroinflammatory processes commonly associated with diverse neurological disorders. This interaction gains significance due to its potential consequences for maintaining the integrity of the BBB. Moreover, the ECS’s engagement in neuroprotective functions may indirectly influence BBB performance by counteracting oxidative stress and inflammatory processes, recognized contributors to BBB impairment. In this manner, the ECS might contribute to sustaining BBB integrity and bolstering overall cerebral well-being. Furthermore, the ECS exhibits implications in the intricate phenomenon of neurovascular coupling, governing the intricate interplay between neuronal activity and blood flow regulation that holds pivotal importance in ensuring appropriate perfusion during heightened neural demands. Perturbations in cerebral blood flow can wield effects on BBB permeability, implicating the ECS in the potential coordination of neuronal activity, blood flow dynamics, and BBB functionality [49].

## 6. A Note about Sleep, the Glymphatic System, and Endocannabinoids

Endocannabinoids show circadian fluctuations in healthy humans assessed by measurement of plasma AEA and 2-AG levels, with the highest AEA plasma levels occurring upon waking and the lowest just before sleep onset [33,34,35,36,37]. Direct activation of CB1R augmented NREM3 (a phase in which the GS and AQP4 are most active) by stabilizing and increasing the duration of individual bouts, while blockade of CB1R using antagonist AM281 fragmented NREM3.

Also, the observation that CSF clearance within the brain parenchyma is compromised in mice lacking AQP4 underscores the potentially pivotal role of AQP4 in orchestrating CSF homeostasis. Additionally, the absence of AQP4 erases the diurnal variation in glymphatic drainage rates [50].

Notably, the perturbation of the glymphatic drainage, as evidenced by sleep deprivation, correlates with heightened Aβ accumulation, increased levels of ISF tau and CSF tau and α-synuclein in human subjects [51,52,53,54]. This underlines the potential of circadian rhythm disturbances as risk factors in neurodegenerative disorders and emphasizes the importance of the inward flow of CSF through AQP4 channels which occurs during NREM sleep, as reduced expression of AQP4 has been documented both in patient populations and animal models of neurodegenerative diseases.

In addition, sleep deprivation activates nuclear factor-kappa B (NF-κB), a transcription factor that plays a central role in regulating immune and inflammatory responses, resulting in an increased production of pro-inflammatory cytokines, such as interleukin-6 (IL-6) and tumor necrosis factor-alpha (TNFα). Also, oxidative stress caused by sleep deprivation is characterized by an imbalance between the production of reactive oxygen species (ROS) and the body’s ability to neutralize them. Oxidative stress can trigger more inflammation. Lastly, sleep deprivation can lead to the activation of microglia and increase the expression of adhesion molecules on endothelial cells, facilitating the recruitment of immune cells to the CNS and promoting neuroinflammation [55,56].

## 7. Potential Interactions between the GS, the BBB, and the ECS

Alteration of GS drainage function and disruption of BBB integrity might contribute to the impediment of effective clearance mechanisms for toxic proteins (waste products) in the context of neurodegenerative conditions.

The ECS is undoubtedly a central regulatory force in the functioning of both the GS and the BBB, with these systems in turn exerting reciprocal influences on endocannabinoid signaling. This interwoven relationship underscores the interconnected nature of the ECS, GS, and BBB, collectively serving as integral components in the maintenance of brain equilibrium and the orchestration of neuroinflammatory processes.

Neuroinflammation plays a central role in the context of neurodegenerative disorders. Activation of the ECS offers the potential to ameliorate neuroinflammation by attenuating the release of pro-inflammatory agents, dampening glial activation, and stimulating anti-inflammatory cascades. Furthermore, the ECS exerts regulatory influence over the GS, impacting essential components governing waste elimination and the dynamics of ISF.

Cannabinoids, encompassing endocannabinoids and phytocannabinoids, may function as signaling molecules within the GS, orchestrating fluid dynamics and waste clearance processes. For example, the activation of CB1R might confer protection against excitotoxicity and the initiation of reparative mechanisms following neuronal damage, while CB2R activation could potentially influence the expansion and contraction of perivascular spaces, thereby facilitating the elimination of metabolic waste through the glymphatic pathway [57] (Figure 2). Similarly, these receptors could influence the expression and stability of tight junction proteins, consequently regulating BBB permeability. Perturbation of the ECS may precipitate dysfunction in both the BBB and GS, culminating in neuroinflammatory processes [49]. The GS’s proficiency in waste removal and the elimination of inflammatory mediators could potentially mitigate neuroinflammation, indirectly influencing the ECS by cultivating a conducive milieu for its optimal operation.

Furthermore, both the GS and the BBB likely partake in the reciprocal modulation of endocannabinoid signaling in the brain. The GS’s efficient waste removal and ISF dynamics might significantly affect the availability and clearance of endocannabinoids within the cerebral milieu. By facilitating the removal of surplus endocannabinoids, the GS may contribute to upholding an appropriate endocannabinoid tone, thereby preventing aberrant signaling and the emergence of possible neuroinflammation. Concurrently, the BBB operates as a selective barricade for the transportation of cannabinoids, managing their entry into and exit from the brain.

Perturbations in BBB permeability and transport mechanisms could potentially alter the availability and duration of cannabinoid signaling within the cerebral realm.

Scientific evidence has effectively showcased the potential of cannabinoids, both endogenous and exogenous in origin (derived from cannabis or synthesized), in mitigating symptomatic manifestations associated with diverse neurodegenerative ailments encompassing MS, HD, PD, AD, and ALS. This influence may stem from their effects on the GS, facilitating the clearance of neurotoxic substances and protein aggregates, regulating neuroinflammatory responses, and upholding cerebral equilibrium (Figure 2). Increased expression of cannabinoid receptors, particularly CB2R, has been noted in human brain tissues afflicted by ALS, MS, and AD. The interplay between the BBB and potential shifts in the expression of cannabinoid receptors within cerebral endothelium has been the subject of investigation.

These insights collectively suggest that a compromised or dysregulated ECS could potentially contribute to the symptomatology observed in these conditions through direct modulation of the GS and the BBB.

In the subsequent segments, we will explore potential interactions between the ECS and the GS, delving into their ramifications for cerebral well-being and their relevance within the context of neurodegenerative diseases.

### 7.1. Alzheimer’s Disease

AD is the most prevalent form of dementia, a debilitating neurodegenerative disorder that primarily affects individuals over the age of 65. The etiology of AD involves a combination of genetic and idiopathic factors, leading to significant neuronal atrophy in the cerebral cortex, hippocampus, and glutamatergic neurons. This results in the extracellular accumulation of Aβ protein in plaques and the formation of intracellular hyperphosphorylated Tau protein tangles, leading to neurodegeneration [58].

In AD, the deposition of Aβ plaques and tau protein aggregates hampers the proper functioning of the GS. This leads to reduced waste clearance and the accumulation of toxic proteins, contributing to neuroinflammation and neuronal damage. The compromised GS may also contribute to the spread of pathological proteins in a prion-like manner, further propagating disease pathology [58].

Modifications in the expression and operational efficacy of CB1R have been documented in the cerebral tissues of individuals afflicted with AD and in experimental models mimicking AD. Intriguingly, a congruent decline in CB1R levels has been identified in the brains of both patients with AD and AD animal models during the advanced stages of the condition. This contrasts with the augmented CB1R expression observed during the asymptomatic phases [59,60,61].

The neuroprotective mechanism of the ECS involves diverse pathways. Activation of CB1R, responsible for modulating the release of excitatory neurotransmitters from presynaptic neurons, offers protection against excitotoxicity and promotes neurogenesis. In parallel, the activation of CB2R diminishes oxidative stress and curtails neuroinflammation by suppressing microglial activation and regulating the production of inflammatory mediators.

Emerging research suggests that the ECS’s positive impact on mitigating Aβ brain accumulation might be attributed to an augmented transport of Aβ out of the brain [62]. This effect seems to stem from an increased expression of the low-density lipoprotein receptor-related protein 1 (LRP1), which is known to participate in the transport of Aβ from the brain to the blood. These investigations suggest that the ECS plays a part in eliminating Aβ from the brain to the periphery, potentially elucidating its influence on Aβ brain accumulation and the pathophysiology of AD [63,64].

The facilitation of Aβ transit across the BBB through the ECS, aligns with the GS’s ability to modulate the exchange of CSF and ISF, influencing waste elimination pathways. It is conceivable that the cannabinoid-induced enhancement of Aβ clearance from the brain to the periphery, in collaboration with the GS, aids in maintaining a balanced Aβ homeostasis. The GS’s proficiency in the removal of solutes and waste products, potentially including Aβ, could synergistically augment the effects of the ECS, thereby contributing to the alleviation of Aβ burden and potentially impacting AD progression [65].

### 7.2. Multiple Sclerosis

The etiology of MS is intricate and encompasses a convergence of genetic, environmental, and immunologic components that culminate in immune-mediated impairment of the CNS. The prevailing hypothesis posits MS as an autoimmune disorder wherein the immune system erroneously targets the myelin sheath enveloping nerve fibers within the CNS. This demyelination perturbs neural signal propagation, resulting in an array of neurological manifestations.

Central to MS pathogenesis is the perceived pivotal involvement of immune cells, notably T cells. Susceptible individuals, genetically predisposed, can encounter environmental triggers like viral infections, precipitating the activation of autoreactive T cells. These T cells traverse the blood-brain barrier and infiltrate the CNS, where they misconstrue myelin proteins as antigens, inciting an immune reaction [66]. This immune response activates immune cells, including macrophages and microglia, which discharge pro-inflammatory cytokines, chemokines, and other agents fostering inflammation and detriment to myelin and nerve cells. This inflammatory cascade can give rise to demyelinated plaques and axonal loss, the elongated projections of nerve cells accountable for transmitting electrical impulses [66,67].

Subsequently, the CNS endeavors to effect repair and remyelination, albeit often incomplete, potentially resulting in the formation of scar tissue (gliosis). The recurring cycles of inflammation, demyelination, and repair attempts contribute to the distinctive relapsing-remitting MS pattern, wherein symptoms exacerbate during relapses and partially alleviate during remissions [67].

As the condition progresses, for some patients, the extent of inflammation and damage may broaden, fostering the emergence of secondary progressive MS characterized by gradual symptom exacerbation and escalating disability.

While the primary mechanisms of MS focus on immune-mediated damage and demyelination, the GS’s potential implications cannot be overlooked. The GS, responsible for the clearance of interstitial waste products from the CNS, could intersect with the immune responses observed in MS. Inflammation and immune cell infiltration, hallmarks of MS, might impact glymphatic function, potentially influencing the removal of toxic molecules and immune mediators from the CNS environment. Conversely, compromised glymphatic function due to factors like disrupted AQP4 channels or altered interstitial fluid dynamics could affect waste clearance and contribute to the build-up of neuroinflammatory agents. The GS’s dysfunction may also affect the clearance of immune cells, inflammatory molecules, and myelin debris from the CNS. This impaired clearance can perpetuate chronic inflammation and neurodegeneration, contributing to the progression of the disease.

In addition to the multifaceted interactions within the realm of MS pathogenesis, exploring the potential involvement of the ECS adds another layer of complexity and intrigue.

Activation of cannabinoid receptors has been found to have immunomodulatory and anti-inflammatory effects in experimental models of MS. The endocannabinoids themselves have been shown to exert neuroprotective and anti-inflammatory actions, potentially limiting the damage caused by the immune response in MS. Synthetic cannabinoid agonists like HU210 or WIN 55212-2 have been found to protect oligodendrocytes from apoptosis-induced by trophic factor deprivation, acting on both CB1R and CB2R. These agonists also suppress the production of inflammatory molecules, including IL-1β, TNFα, and NO, by astrocytes and microglial cells. Additionally, they enhance the release of anti-inflammatory cytokines such as IL-4, IL-10, IL-6, and interleukin-1 receptor antagonists (IL-1ra) [68].

Cannabinoid administration offers a potential strategy to halt the progression of MS by targeting both the GS and the ECS. By activating cannabinoid receptors, cannabinoids could promote anti-inflammatory responses and enhance waste clearance through the GS. They might attenuate the activation of immune cells that contribute to neuroinflammation and demyelination, while also facilitating the removal of neurotoxic substances through the glymphatic pathway. This dual action could collectively alleviate the inflammatory burden on the CNS and potentially slow down disease progression [69,70].

### 7.3. Huntington’s Disease

HD is an autosomal dominant inherited neurodegenerative disorder characterized by involuntary choreiform movements, cognitive impairment, metabolic abnormalities, and a relentlessly progressive course culminating in death 10–25 years after onset. The genetic basis of HD is the expansion of a CAG trinucleotide repeated within the Huntingtin (HTT) gene, resulting in the production of HTT protein containing an expanded glutamine tract. This altered peptide is resistant to normal cellular processes of protein turnover and “aggregates” or “inclusions” of the aberrant protein accumulate within neurons in HD brain regions.

Compelling evidence underscores the association between HD and the ECS. One of the earliest neurochemical changes in HD involves the depletion of CB1R binding in the basal ganglia [71,72]. In post-mortem HD brains, binding investigations have elucidated a specific and substantial reduction in CB receptor presence, notably within the substantia nigra [73]. Furthermore, animal models replicating HD conditions have revealed impairment in the ECS, evident through diminished CB1R expression and altered endocannabinoid tissue concentrations [74,75]. These observations suggest a potential correlation between lowered CB1R expression and the severity or progression of the disease. In addition, CB2Rs present in the striatum increase prior to symptom onset [76].

An in-depth exploration of the intricate relationship between HD and the ECS will reveal more intriguing insights. Analysis of lymphocyte preparations of patients with HD revealed that AEA levels were six-fold higher than those of control patients [77]. This noteworthy phenomenon can be attributed to the inhibition of FAAH function, a key enzyme responsible for AEA metabolism. This particular alteration within the ECS offers a unique perspective into the underlying molecular intricacies of HD, hinting at the potential impact of endocannabinoid signaling on the disease’s pathophysiology. The heightened AEA levels could potentially contribute to the modulation of various cellular processes and neuroinflammatory responses that are intricately tied to the progression of HD.

The dysregulation of the ECS in HD may impact various disease-related processes, including neuroinflammation, excitotoxicity, and oxidative stress. Modulation of the ECS through cannabinoid receptor agonists or inhibitors of endocannabinoid breakdown enzymes has shown potential therapeutic effects in preclinical models of HD, such as reducing neuroinflammation and improving motor function. Dysfunction in the GS could potentially contribute to the accumulation of mutant huntingtin protein and exacerbate neuronal damage in Huntington’s disease.

### 7.4. Parkinson’s Disease

PD is a neurodegenerative disorder of the CNS characterized by the degeneration of dopaminergic neurons in the substantia nigra. This leads to insufficient production and action of dopamine, resulting in decreased stimulation of the motor cortex by the basal ganglia. The disease is characterized by primary symptoms such as muscle rigidity, tremors, and bradykinesia (slowing of physical movements), as well as secondary symptoms including cognitive dysfunction and subtle language problems [78].

The neuromodulatory effects of the ECS are closely linked to the dopaminergic system, and there exists a reciprocal regulation between these systems. This is evident in the co-localization of CB1 and D1/D2-like receptors in striatal neurons and their complex signaling interactions [79]. For instance, studies have shown that AEA can reduce dopamine release in striatal slice cultures but increase it in the nucleus accumbens in vivo [80,81]. Additionally, activation of D2 receptors has been found to elevate AEA levels in the basal ganglia [82]. It is noteworthy to mention that only a few studies have investigated the levels of endocannabinoids in PD patients, revealing that AEA levels in their cerebrospinal fluid were more than twice that of controls [83,84].

Cannabinoid receptor expression and endocannabinoid dysregulation may contribute to the underlying neuroinflammatory processes, oxidative stress, and neuronal dysfunction associated with the disease, as CB2R agonism exerts neuroprotective effects by decreasing inflammation and microglia activity, inhibiting the release of pro-inflammatory cytokines, and promoting the release of anti-inflammatory cytokines, as well as increasing glutamate uptake [85,86].

Similarly, in PD, α-synuclein aggregates and Lewy bodies can impede glymphatic flow, impairing the removal of waste products and exacerbating neuroinflammation [85]. Additionally, patients with PD, especially those in the later PD stages, have a significantly lower APLS score than early-stage PD, essential tremor patients, and healthy patients [87,88]. The involvement of the GS PD is still being explored, and its specific role in the disease remains to be fully elucidated. Dysfunction in the GS could potentially contribute to the accumulation of toxic proteins, impaired waste clearance, and exacerbation of neuroinflammatory processes observed in PD.

### 7.5. Amyotrophic Lateral Sclerosis

ALS is primarily characterized by the degeneration of motor neurons in the cortex, brainstem, and spinal cord [89,90]. The underlying etiopathological mechanisms primarily involve neuroinflammation, driven by excitotoxicity and oxidative damage to motor neurons.

The BBB plays a significant role in the pathophysiology of ALS because there are specialized interfaces that control de flow of nutrients and ions into the CNS and the removal of waste and other substances. Furthermore, this barrier serves to separate the brain, spinal cord tissue, and CSF from potentially harmful blood-borne elements found in the circulation, such as peripheral leukocytes, red blood cells, and plasma proteins. In cases of ALS, there is a compromise in the integrity of the BBB and BSCB, as supported by data from human studies. This disruption is evident through the leakage of blood components like immunoglobulin G (IgG), fibrin, thrombin, and the constituents of red blood cells, hemoglobin, and hemosiderin, into the brain and spinal cord tissue [49,91,92].

Experimental evidence in the spinal cords of ALS patients has revealed motor neuron damage accompanied by the presence of CB2R-positive microglia/macrophages [91,93].

Treatment of ALS mouse models with Delta-9-tetrahydrocannabinol (D9-THC), a cannabinoid compound, has shown symptomatic improvement when administered before or after the onset of symptoms [93,94]. Studies conducted using ALS mice have demonstrated that activation of CB2Rs has been shown to block microglial activation induced by Aβ, but under other stimuli, CB2R activation has been observed to enhance microglial migration and proliferation. Other studies have shown that the use of CB2R agonists can slow disease progression, even when administered after the onset of the disease. Additionally, one study reported a 56% increase in survival time. Moreover, the analysis of activated microglia in the spinal cords of ALS patients has revealed an increase in CB2R expression. These findings collectively suggest that modulating CB2-mediated processes could potentially alter the progression of ALS and highlight the involvement of the ECS in reducing neuroinflammation, excitotoxicity, and oxidative damage to cells [49,95]. Research has shown that impairment of the GS could lead to reduced clearance of toxic proteins, such as misfolded superoxide dismutase 1 (SOD1), which is associated with familial ALS [90,96,97]. This impaired clearance may contribute to the accumulation of misfolded proteins and subsequent neurodegeneration in ALS. Additionally, studies have suggested a potential interaction between the glymphatic and endocannabinoid systems. Activation of cannabinoid receptors has been shown to modulate inflammation and promote clearance of Aβ plaques, suggesting a potential role for the endocannabinoid system in glymphatic function and waste clearance [98,99].

## 8. Future Directions and Conclusions

The putative correlation between the GS and the ECS holds great promise for advancing our understanding of brain health and disease. Further investigations are warranted to elucidate the precise mechanisms underlying their interconnection and explore potential therapeutic interventions. Manipulation of the endocannabinoid system may offer new strategies to enhance glymphatic clearance, thereby promoting brain health and mitigating the pathological processes associated with neurodegenerative disorders. Continued research in this area is crucial for developing novel treatment approaches, and scientific divulgation is necessary for the development of public policies that improve the quality of life for individuals affected by devastating conditions [100].

Enhancing glymphatic function represents a promising therapeutic avenue for neurological disorders. Modulating sleep patterns, promoting glymphatic activity through pharmacological agents, and targeting molecular pathways involved in the GS could potentially facilitate waste clearance and mitigate neurodegenerative processes. Furthermore, the GS’s role in drug delivery to the brain highlights its relevance in optimizing drug efficacy and reducing toxicity.

Despite significant advances in glymphatic research, several aspects require further investigation. These include elucidating the precise mechanisms governing glymphatic clearance, determining the interplay between the GS and other waste clearance pathways, and exploring the relationship between glymphatic dysfunction and neurovascular diseases. Advanced imaging techniques, animal models, and innovative experimental approaches will be instrumental in advancing our understanding of the GS.

The ECS has garnered significant interest as a potential therapeutic target for a wide range of medical conditions. Pharmacological modulation of the ECS through the use of cannabinoid-based therapeutics, including synthetic cannabinoids, cannabinoid receptor agonists, and inhibitors of endocannabinoid degradation enzymes, holds promise for the development of novel treatments.

The ECS is a complex and versatile signaling network that regulates various physiological processes throughout the body. Its involvement in pain modulation, inflammation, neuronal plasticity, appetite regulation, and mental health disorders highlights its significance in human health and disease. The ECS represents a promising target for therapeutic interventions, with the potential to revolutionize the treatment of a wide range of medical conditions.

The correlation between the ECS and the BBB has significant therapeutic implications. Targeting the ECS could potentially modulate BBB permeability and improve drug delivery to the brain. Additionally, modulation of the ECS may have therapeutic potential in neuroinflammatory conditions and neurodegenerative diseases by regulating immune responses and protecting BBB integrity. The ECS, through its modulation of the BBB integrity, neuroinflammation, and immune responses, plays a crucial role in the regulation of brain health.

The ECS and the GS are interconnected and mutually regulate each other’s functions. Activation of the ECS influences glymphatic function and waste clearance, while the GS modulates the availability and metabolism of endocannabinoids, impacting their signaling pathways and neuroinflammatory responses. Further research is necessary to elucidate the molecular mechanisms and signaling pathways underlying this bidirectional regulation, shedding light on the therapeutic potential of targeting the endocannabinoid-glymphatic axis in neurodegenerative diseases.

## Figures and Tables

**Figure 1 ijms-24-17458-f001:**
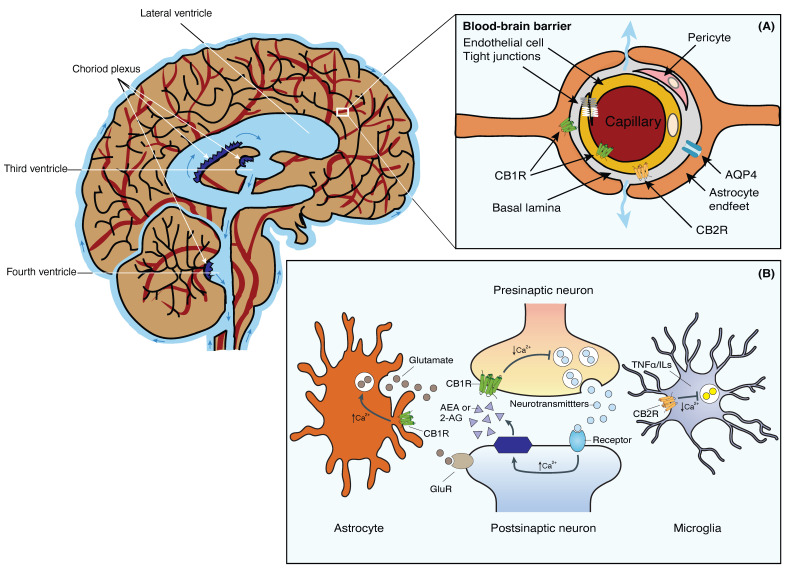
Schematic representation of the brain’s fluid compartments and endocannabinoid receptors in the nervous and glia system. Main: The cerebral ventricles are a system of interconnected, fluid-filled cavities within the brain that contribute to cerebrospinal fluid (CSF) production and circulation. The four ventricles include the paired lateral ventricles, the third ventricle, and the fourth ventricle, connected by the cerebral aqueduct. These ventricles are lined with ependymal cells and house the choroid plexus, a specialized structure responsible for CSF secretion. Inset (**A**): Tight junctions between the blood endothelial cells constitute the BBB, restricting macromolecules from moving freely from the blood into the brain parenchyma. Fluid and solutes diffuse into the brain parenchyma from the perivascular space located between endothelial cells and astrocytic endfeet that expresses AQP4 and CB1R. CB1Rs are primarily located on the luminal side of the BBB endothelium. CB2Rs, on the other hand, are located on the abluminal side of the BBB. Inset (**B**): 2-AG or AEA are synthesized from phospholipids on demand. Activation of presynaptic CB1R negatively modulates cell calcium influx and the release of neurotransmitters in neurons. Stimulation of CB1R in astrocytes positively modulates calcium influx and glutamate release. Activation of CB2R in microglia negatively affects the release of TNFα and ILs. Abbreviations: AQP4, Aquaporin-4 water channel; BBB, blood-brain barrier; CB1R, Cannabinoid receptor 1; CB2R, Cannabinoid receptor 2; CSF, cerebrospinal fluid; 2-AG, 2-acylglycerol; AEA, anandamide; GluR, glutamate receptors; ILs, interleukins; TNFα, tumor necrosis factor-α. This figure was made in Adobe Illustrator 2020, modified from Jessen et al., 2015 [3].

**Figure 2 ijms-24-17458-f002:**
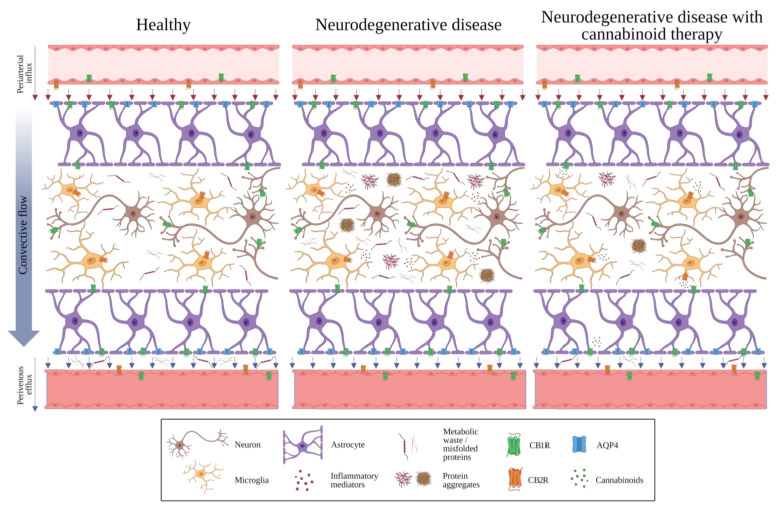
Schematic representation of the blood-brain barrier, the glymphatic system, and the endocannabinoid system under 3 different conditions. Healthy: Interconnected regulation of CNS homeostasis under normal conditions. The BBB forms a protective barrier between the bloodstream and the brain, comprising tightly packed endothelial cells, pericytes, and astrocytic endfeet. This selective barrier strictly regulates the passage of molecules into and out of the brain, maintaining optimal microenvironmental conditions. The GS operates as a CSF and ISF drainage network where specialized perivascular spaces enable the efficient clearance of metabolic waste products and neurotoxic substances, ensuring effective waste management and maintaining brain homeostasis. Cannabinoid receptors (CB1R and CB2R) are strategically located on neurons, astrocytes, and microglia, allowing for versatile regulation of CNS functions. Neurodegenerative disease: The GS experiences impaired drainage and clearance mechanisms, leading to the accumulation of metabolic waste, toxic proteins, and inflammatory mediators. Neurodegenerative disease with cannabinoid therapy: In neurodegenerative conditions under cannabinoid treatment, the GS showcases improved drainage and clearance mechanisms, promoting the efficient removal of metabolic waste and toxic proteins. Simultaneously, the ECS reflects a restored balance and a reduction of inflammatory mediator synthesis. Abbreviations: AQP4, Aquaporin-4 water channel; CB1R, Cannabinoid receptor 1; CB2R, Cannabinoid receptor 2. This figure was created with BioRender.com, agreement number YJ265GYS33, 28 November 2023.

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
