# Peer review of "Interplay between the Glymphatic System and the Endocannabinoid System: Implications for Brain Health and Disease"

_ijms, 2023, doi:10.3390/ijms242417458_

Round 1
Reviewer 1 Report
Comments and Suggestions for Authors
This is a well written review on the Interplay between the GS and the Endocannabinoid system. I would recommend that you spell out GS inthe title. The Glymphatic System is well defined and described. This is also true of the Endoconnabinoid System. The authors did an excelleent job of tying the references to each subject heading and explaining in detail how each system works and are interconnected. I must state that I was most impressed on how extensive you tied the system to each. You clearly showed the relationship of the two system to the most common disorders (Alzheimer's, Multiple Sclerosis, Parkinson'sHuntington's, and Amyotrophic lateral sclerosis. The authors are very perceptive on the future need for more scientific work needs to done on the subject matter. Please correct me If I missed it ,but I did not find any author in the references that wrote this review. You done an excellent job on a very extensive and comprehensive references for the paper.
Author Response
We would like to express our sincere gratitude for your thoughtful and thorough review of our paper, "Interplay between the Glymphatic System and the Endocannabinoid System: Implications for Brain Health and Disease."
We are genuinely appreciative of the time and effort you dedicated to this review, as well as for your thoughtful comments.
We eliminated all abbreviations from the title and abstract.
Reviewer 2 Report
Comments and Suggestions for Authors
Major comments:
The figure is missing.
The introduction is missing.
All the subheadings are incomplete.
A mechanism and connection figures are needed.
A figure is needed in each neurodegenerative disease part.
Minor comments:
Some of the basic information should be followed by the authors while preparing the manuscript, like
The title should not contain abbreviations.
What is GS in the abstract?
The manuscript is not in the template; line numbers are missing.
References are not in the specified format.
The font is not uniform.
Author Response
We appreciate the constructive feedback and insightful comments, which have proven invaluable in refining the quality and depth of our work "Interplay between the Glymphatic System and the Endocannabinoid System: Implications for Brain Health and Disease." We are genuinely appreciative of the time and effort you dedicated to this review.
Changes made in the manuscript are highlighted in color red.
Here are the responses to the reviewer's comments:
1. The figure is missing.
We apologize for this, we submitted figure 1 and figure 2 to complement the manuscript.
- The introduction is missing.
We added an introduction to the manuscript.
- All the subheadings are incomplete.
We modified the subheadings.
- A mechanism and connection figures are needed.
We included figure 1 and figure 2 to the manuscript.
- A figure is needed in each neurodegenerative disease part.
We added figure 2, which includes a general mechanism for all neurodegenerative diseases.
- The title should not contain abbreviations.
We apologize for this error, we modified the title.
- What is GS in the abstract?
We apologize for this error, we eliminated all abbreviations in the title and abstract. GS stands for Glymphatic System.
- The manuscript is not in the template; line numbers are missing.
We apologize for this inconvenient, we believe that this is a technical error related to the submission system.
- References are not in the specified format.
We modified the reference style to match the specified format.
- The font is not uniform.
Now the entire manuscript is with the same font.
Round 2
Reviewer 2 Report
Comments and Suggestions for Authors
Still, the manuscript has multiple fonts, not uniform.
Figures are missing, I cant find the figures in both manuscript and supplement files.
Author Response
Dear reviewer,
We have taken the necessary steps to address the concerns you raised. Specifically, regarding the issue of multiple fonts and missing figures. However, it appears that there may be an issue with the submission platform. We have meticulously reviewed both the main manuscript and supplement files to verify the presence of all figures. We uploaded a single PDF file containing the manuscript and figures (at the end of the manuscript).
Best regards
